# Bioengineered Approaches for Esophageal Regeneration: Advancing Esophageal Cancer Therapy

**DOI:** 10.3390/bioengineering12050479

**Published:** 2025-04-30

**Authors:** Jae-Seok Kim, Hyoryung Nam, Eun Chae Kim, Hun-Jin Jeong, Seung-Jae Lee

**Affiliations:** 1Department of Mechanical Engineering, College of Engineering, Wonkwang University, 460 Iksandae-ro, Iksan 54538, Republic of Korea; jaeseokkim311@gmail.com (J.-S.K.); kimeunchae1008@naver.com (E.C.K.); 2Department of Biomedical Engineering, School of Medicine, Daegu Catholic University, 33 Duryugongwon-ro 17-gil, Nam-gu, Daegu 42472, Republic of Korea; ryung@cu.ac.kr; 3College of Dental Medicine, Columbia University Irving Medical Center, 630 W. 168th St., VC12-212A, New York, NY 10032, USA; 4Division of Mechanical Engineering, College of Engineering, Wonkwang University, 460 Iksandae-ro, Iksan 54538, Republic of Korea; 5MECHABIO Group, Wonkwang University, 460 Iksandae-ro, Iksan 54538, Republic of Korea; 6Advanced Bio-Convergence Research Center, Wonkwang University, 460 Iksandae-ro, Iksan 54538, Republic of Korea

**Keywords:** esophageal cancer, esophageal reconstruction, tissue engineering, tubular-type scaffolds, patch-type scaffolds, animal models

## Abstract

Esophageal cancer (EC) is the eighth leading cause of cancer-related deaths globally, largely due to its late-stage diagnosis and aggressive progression. Esophagectomy remains the primary treatment, typically requiring organ-based reconstruction techniques such as gastric pull-up or colonic interposition. However, these reconstruction methods often lead to severe complications, significantly reducing the quality of life of patients. To address these limitations, tissue engineering has emerged as a promising alternative, offering bioengineered patch-type and tubular-type scaffolds designed to restore both structural integrity and functional regeneration. Recent advancements in three-dimensional (3D) biofabrication—including 3D bioprinting, electrospinning, and other cutting-edge techniques—have facilitated the development of patient-specific constructs with improved biocompatibility. Despite significant advancements, critical challenges persist in achieving mechanical durability, multilayered cellular organization, and physiological resilience post-transplantation. Ongoing research continues to address these limitations and enhance clinical applicability. Therefore, this review aims to examine recent advancements in esophageal tissue engineering, with a focus on key biofabrication techniques, preclinical animal models, and the major translational challenges that must be addressed for successful clinical application.

## 1. Introduction

Esophageal tissue is a crucial component of the digestive system, primarily responsible for transporting food and liquids to the stomach. The human body has a single esophagus characterized by a distinct hierarchical cellular structure. This includes the muscularis layer, which facilitates the movement of ingested substances through coordinated muscle contractions, and the mucosal layer, which protects against digestive enzymes and stomach acid [1]. Unlike other parts of the gastrointestinal tract, the esophagus lacks a serosal layer, making it more susceptible to perforation and impaired healing following injury or surgical intervention [2]. Owing to its structural complexity and essential function, esophageal damage—whether caused by congenital defects, trauma, or malignancy—presents significant clinical challenges [3,4].

Esophageal cancer (EC), which includes esophageal adenocarcinoma (EAC) and esophageal squamous cell carcinoma (ESCC), is a common malignancy with a high mortality rate globally. Unfortunately, EC exhibits clinical characteristics such as rapid progression, early metastasis, and poor survival rates [5,6]. The primary treatment for EC is surgical removal of the affected area, followed by reconstruction using another organ, most commonly the stomach or intestine. However, esophagectomy is considered a high-risk surgical procedure with a significant possibility of serious postoperative complications [5]. Conventional reconstruction techniques, such as gastric pull-up and colonic interposition, do not restore native esophageal peristalsis, leading to persistent swallowing difficulties and a reduced quality of life for patients [7].

To address these limitations, tissue engineering approaches have emerged as promising strategies for creating transplantable esophageal constructs. Unlike conventional grafts, which primarily restore anatomical continuity, tissue-engineered esophageal scaffolds are designed to facilitate functional regeneration by incorporating biocompatible materials, growth factors, and patient-derived cells. Advanced three-dimensional (3D) biofabrication techniques, including 3D bioprinting, electrospinning, and decellularized extracellular matrix (dECM) scaffolds, enable the development of patient-specific constructs that closely replicate the structural and biomechanical properties of the native esophagus [8]. Current leading tissue engineering strategies include patch-type scaffolds for repairing localized esophageal defects and tubular scaffolds for full-thickness circumferential reconstruction [9]. Promising outcomes have been reported with advanced tissue engineering fabrication methods that integrate specific growth factors and stem or progenitor cells to promote the development of epithelial and smooth muscle cell layers [10,11,12,13]. However, significant challenges remain, including inadequate mechanical strength to support functional movement and the complexity of achieving multilayered cellular structures. Furthermore, considering the unique physiological environment of esophageal tissue—which is exposed to unsterilized food following transplantation—tissue-engineered artificial esophageal structures must account for both functional recovery, including appropriate mechanical properties and multi-cellular components, and potential physiological factors contributing to failure.

Accordingly, this study offers a comprehensive overview of recent progress in esophageal tissue engineering, with a particular focus on technological innovation and the challenges currently facing the field (Figure 1). It summarizes key research trends and technical directions grounded in various reconstruction strategies and integrates recent findings from animal model studies to evaluate their strengths and limitations. Drawing on these insights, this study outlines the major challenges in esophageal tissue engineering and presents a cohesive perspective on future opportunities and directions for advancement.

## 2. Clinical Overview of Esophageal Cancer

### 2.1. Comprehensive Overview of Esophageal Cancer

EC has a significantly low 5-year survival rate, making it the eighth most common cancer worldwide with a high mortality rate [14]. Additionally, the disease is the sixth leading cause of cancer-related deaths globally, highlighting its significant public health burden [15]. EC is primarily classified into two major histological subtypes: EAC and ESCC. While these subtypes differ in their pathogenesis and anatomical distributions, they present similar clinical challenges, including poor prognosis, aggressive tumor progression, and a high likelihood of metastasis. Consequently, early detection and timely intervention are essential for improving patient outcomes [5,16].

EAC predominantly develops in the lower esophagus and is closely associated with Barrett’s esophagus (BE), a precancerous condition caused by chronic damage to the esophageal lining. While this subtype typically progresses slowly, the risk of malignant transformation increases significantly without timely intervention [17]. In contrast, ESCC generally develops in the middle or upper thirds of the esophagus and is highly aggressive, exhibiting rapid local invasion and early lymphatic spread. These factors often lead to late-stage diagnoses and poorer clinical outcomes [18].

Clinically, EC often presents with non-specific, gradually worsening symptoms, leading to delayed diagnosis. Dysphagia (difficulty swallowing) is the most common initial symptom and progressively worsens as the tumor obstructs the esophageal lumen [19]. Other symptoms may include significant weight loss, chest pain, odynophagia (painful swallowing), and, in advanced stages, hoarseness due to recurrent laryngeal nerve involvement [20]. Tumor invasion into the trachea or bronchi can cause respiratory complications, such as chronic cough and aspiration [21]. Nutritional deficiencies often occur due to impaired swallowing and reduced oral intake, leading to weakened immune function and overall health deterioration [22]. Beyond the physical complications, EC and its treatment impose significant psychological distress, further diminishing the quality of life of the patient [23].

### 2.2. Prevalence

EC represents a major global health concern, with its prevalence varying significantly across regions. Annually, approximately 572,034 new cases are diagnosed, and 508,585 deaths occur worldwide, accounting for approximately 5.35% of all cancer-related mortality [14].

The incidence rate of EC varies across the Americas, Europe, Asia, and Africa due to region-specific genetic, environmental, and socioeconomic factors. In Western countries, EAC has been rising significantly, primarily due to its strong association with chronic acid reflux, dietary patterns, and obesity. Additionally, ESCC is highly prevalent in Asian and African regions, largely due to lifestyle factors such as smoking, alcohol consumption, and dietary patterns. Additionally, sex differences are evident, with men, particularly those ≥ 50 years, exhibiting a significantly higher incidence of EC than women [15]. These prevalence patterns serve as crucial indicators for developing future prevention and treatment strategies.

### 2.3. Key Risk Factors for Esophageal Cancer

EC develops due to a complex interaction of lifestyle, environmental, and genetic factors, each playing a different role in the pathogenesis of its two primary subtypes: EAC and ESCC [5,18,24,25].

Among lifestyle-related risk factors, smoking and excessive alcohol consumption are the primary contributors to ESCC. Carcinogens in tobacco smoke, including polycyclic aromatic hydrocarbons and nitrosamines, along with alcohol metabolites such as acetaldehyde, induce chronic irritation and cellular damage to the esophageal lining. These harmful agents often act synergistically, significantly increasing the risk of ESCC development [24,26,27,28,29,30].

In contrast, EAC is strongly associated with gastroesophageal reflux disease (GERD) and its complication, BE [31,32,33]. Chronic acid reflux in GERD causes persistent inflammation of the lower esophagus, leading to the replacement of normal squamous epithelium with columnar epithelium, a characteristic of BE. This metaplastic transformation significantly increases the risk of malignant progression to EAC. Obesity, particularly central (abdominal) obesity, exacerbates GERD by increasing intra-abdominal pressure, further elevating the risk of BE and, consequently, EAC. Additionally, obesity-induced systemic inflammation creates a pro-carcinogenic environment that facilitates tumor initiation and progression [34].

Dietary patterns play a significant role in ESCC. High consumption of processed meats, red meats, and excessively hot beverages has been associated with an increased risk of developing ESCC, primarily due to exposure to nitrosamines and chronic thermal injury to the esophageal lining. Conversely, diets rich in fruits and vegetables, which provide high levels of antioxidants and essential micronutrients, may offer a protective effect against ESCC. The dietary influence on EAC development remains less well-defined but may be indirectly influenced by its effect on GERD, as high-fat, low-fiber diets can worsen reflux symptoms [35,36].

Genetic predisposition plays a crucial role in influencing the susceptibility of an individual to EC. Mutations in tumor suppressor genes, such as TP53 and CDKN2A, have been implicated in the pathogenesis of both EAC and ESCC. Additionally, hereditary syndromes, such as Lynch syndrome and familial BE, are associated with an increased risk of EAC. In contrast, genetic polymorphisms affecting alcohol-metabolizing enzymes may contribute to a higher susceptibility to ESCC in certain populations [37,38,39,40]. Recent studies have identified several genes, such as TP53, SMAD4, ARID1A, and PIK3CA, that are significantly related to BE and EAC [41,42].

## 3. Clinical Strategies and Limitations

### 3.1. Current Treatment Strategies

The management of EC is primarily determined by disease stage rather than histological subtype, with treatment strategies adapted based on tumor progression and metastatic spread. The Tumor-Node-Metastasis (TNM) staging system, developed by the American Joint Committee on Cancer (AJCC), is the standard framework for classifying EC. In this system, T denotes the size and local invasion of the primary tumor, N indicates the degree of regional lymph node involvement, and M represents the presence of distant metastasis (Table 1).

In early-stage EC, where malignant cells are limited to the mucosa or submucosa without lymph node involvement, endoscopic therapies are often the first-line approach. However, if the tumor has infiltrated deeper into the submucosa, surgical intervention is often required due to the increased risk of lymph node metastasis [44,45,46,47]. Endoscopic mucosal resection (EMR) and endoscopic submucosal dissection (ESD) facilitate the localized removal of superficial lesions while preserving the esophagus, offering favorable outcomes with lower morbidity compared to surgical resection [48]. When endoscopic approaches are inadequate, such as in cases of deeper tumor invasion or poor histological differentiation, esophagectomy remains the standard treatment. However, early-stage treatments have limitations, including the risk of local recurrence following endoscopic procedures, necessitating close follow-up, along with perioperative risks associated with esophagectomy, such as anastomotic leaks and pulmonary complications [49].

In more advanced but respectable cases, the tumor infiltrates deeper layers, such as the muscularis propria or adventitia, and may involve regional lymph nodes without evidence of distant metastasis. Surgical resection remains the primary treatment option; however, when the risk of lymph node metastasis is high, neoadjuvant therapy is recommended to improve surgical outcomes and reduce the risk of recurrence [45,46,47,50]. The CROSS trial demonstrated significant survival benefits of neoadjuvant chemoradiotherapy followed by surgery compared to surgery alone, particularly in patients with squamous cell carcinoma, leading to its widespread adoption [51]. However, these approaches still pose significant risks, including high morbidity rates (30–50%), postoperative complications such as anastomotic leaks and strictures, and suboptimal functional outcomes, including dysphagia and gastroesophageal reflux [49]. Additionally, a subset of patients exhibits inadequate responses to neoadjuvant therapy, potentially leading to incomplete tumor resection and poorer prognoses [51].

Overall, despite the availability of stage-specific treatment options for EC, significant challenges persist. High recurrence rates, limited survival benefits, and significant treatment-related morbidity highlight the urgent need for alternative strategies.

### 3.2. Surgical Approaches & Limitations

Current treatment strategies for EC primarily include surgical resection, chemotherapy, and radiation therapy, with the chosen strategy largely dependent on the disease stage [52]. Among these, surgical resection remains the most definitive option, particularly for early-stage and locally advanced cases. The primary objective of esophagectomy is complete tumor removal to minimize the risk of metastasis [53].

Post-esophagectomy reconstruction is essential for restoring gastrointestinal continuity. The most commonly used technique is the gastric pull-up procedure, in which the stomach is mobilized and transposed into the thoracic cavity or cervical region to replace the resected esophagus. This technique is generally preferred due to the rich blood supply and sufficient length of the stomach, which facilitates a relatively straightforward anastomosis [46]. However, when the stomach is unavailable or unsuitable—due to factors such as previous gastric surgeries, inadequate vascular supply, or tumor invasion—alternative reconstruction techniques are employed. These alternatives include colonic interposition (the use of colon segments) or jejunal grafting (the use of a section of the jejunum) for esophageal replacement. Colonic interposition is often preferred due to the sufficient length and resistance of the colon to gastric acid, while a jejunal graft is selected for its peristaltic function and anatomical compatibility with the esophagus [47,54].

Despite their widespread adoption, these reconstructive techniques present significant limitations and are associated with substantial morbidity. Anastomotic leakage is one of the most serious complications, occurring at the surgical connection site between the graft and the remaining esophagus. This leakage can lead to severe infections, mediastinitis, and an increased risk of mortality. Another common complication is anastomotic stricture, in which scar tissue at the anastomotic site causes narrowing, leading to dysphagia (difficulty swallowing) and often requiring repeated endoscopic dilations [46,47].

Functional limitations present significant challenges following esophageal reconstruction. Patients who undergo a gastric pull-up procedure often experience gastroesophageal reflux due to the loss of the lower esophageal sphincter, leading to symptoms such as regurgitation, heartburn, and an increased risk of aspiration pneumonia. Additionally, the lack of coordinated peristalsis in the grafted segment often results in impaired motility, further complicating swallowing and increasing the risk of nutritional deficiencies. In colonic interposition and jejunal grafts, complications such as ischemia, redundancy (leading to food stasis), and dysmotility can adversely affect long-term functional outcomes [46,47,55].

These major complications and functional impairments highlight the limitations of traditional esophageal reconstruction techniques, emphasizing the urgent need for alternative strategies [56,57]. In recent years, tissue engineering has gained traction as a promising solution, particularly with the advancement of 3D biofabricated esophageal scaffolds. These bioengineered constructs are designed to replicate the architecture and biomechanical properties of the native tissue, offering the potential to restore function while minimizing complications associated with conventional grafts. The 3D biofabrication techniques leverage biocompatible materials and patient-derived cells to create a platform for developing customized native tissue constructs that facilitate tissue regeneration and improve functional outcomes [45,58,59,60,61,62,63].

### 3.3. Research Trends and Emerging Needs

The challenges associated with conventional esophageal reconstruction techniques have highlighted the urgent need for alternative surgical approaches. As these limitations have become increasingly evident, research efforts have shifted toward developing innovative, regenerative strategies aimed at addressing the inherent limitations of traditional methods [9,64]. Since the 1990s, significant advancements have been made in esophageal reconstruction and transplantation, with >4000 related articles published in 2023 alone (Figure 2). This growing body of literature highlights a sustained interest in improving esophageal surgery outcomes, driven by the limitations of conventional surgical techniques. Advancements in tissue engineering and regenerative medicine have introduced promising strategies for addressing the complex challenges of esophageal repair and reconstruction [8,9,64].

Tissue engineering has gained prominence as a promising field for reducing the frequency and complexity of surgical interventions while minimizing procedure-related complications (Figure 3). This innovative approach focuses on developing biologically functional tissues that integrate seamlessly with the body of the patient, enhancing both short-term and long-term clinical outcomes. In this context, developing artificial esophagus constructs as viable alternatives to traditional grafts has received increasing attention. One of the most significant advancements in this field is the application of 3D biofabrication techniques, facilitating the precise replication of the complex anatomical and biological structures of the esophagus. Among these techniques, 3D bioprinting has gained significant attention for its ability to generate patient-specific scaffolds that closely replicate the architecture of native esophageal tissue. The precise layering of bioinks composed of cells, biomaterials, and growth factors enables 3D bioprinting to fabricate complex, functional esophageal constructs tailored to individual patient needs. The increasing focus on tissue engineering and regenerative medicine, particularly through advanced 3D biofabrication techniques, highlights a paradigm shift in esophageal reconstruction research. These emerging strategies aim to address the limitations of conventional surgical approaches, offering promising solutions that could significantly improve patient outcomes and quality of life [8,9,64,65].

## 4. Recent Advances in Esophageal Tissue Engineering

### 4.1. 3D Biofabrication

3D biofabrication is an advanced approach in regenerative medicine that enables the precise in vitro recreation of human tissues and organs. In esophageal tissue engineering, this technology is crucial for replicating the complex histological and biological structures of the esophagus, ensuring effective reconstruction and functional restoration. Various fabrication techniques have been developed to generate scaffolds that closely mimic native esophageal tissue. These include 3D bioprinting (e.g., extrusion-, jetting-, vat photopolymerization-based), electrospinning, and mold-based fabrication, each offering distinct advantages in scaffold design and functionality [66,67,68]. They employ various biomaterials, such as hydrogels, synthetic polymers, and decellularized matrices, carefully selected to replicate the biochemical, mechanical, and structural properties of natural esophageal tissues [8].

3D bioprinting is a highly promising biofabrication technique owing to its precision in creating complex, cell-laden structures. This method involves systematically layering bioinks—composed of cells, growth factors, and biomaterials—to generate three-dimensional constructs that support tissue regeneration. A key advantage of 3D bioprinting is its ability to customize esophageal grafts to match the specific anatomical and functional needs of a patient, thereby promoting tissue integration and enhancing functional outcomes [59,60,69,70,71,72]. However, the choice of 3D bioprinting technique—such as extrusion, inkjet, or vat photopolymerization—can significantly influence cell viability and the resulting construct’s mechanical properties [8]. From this perspective, selecting an appropriate 3D printing technique is essential to achieving successful clinical outcomes.

Electrospinning is another widely used method, valued for its ability to mimic the extracellular matrix (ECM). It generates nanofibrous scaffolds that replicate the fibrous architecture of native tissues, providing an optimal environment for cell adhesion, proliferation, and differentiation. Additionally, electrospun scaffolds can be modified using molding techniques to achieve specific structural orientations, enhancing their functionality in tissue regeneration [63,69].

Mold-based fabrication complements other techniques by enabling the production of scaffolds with precise geometric configurations. This method uses biocompatible materials shaped within customized molds to produce esophageal scaffolds with defined architectures, ensuring structural integrity and biological functionality [73,74].

A comparative summary of the three scaffold fabrication techniques—3D bioprinting, electrospinning, and mold-based fabrication—is presented in Table 2, highlighting their core principles, as well as esophagus-specific strengths and limitations.

In esophageal tissue engineering, biofabricated scaffolds are generally classified into patch-type and tubular-type structures based on their clinical application. Patch-type scaffolds are designed for partial-thickness esophageal defects, repairing only a segment of the esophageal wall. In contrast, tubular-type scaffolds are engineered for full-thickness circumferential reconstructions, crucial for extensive esophageal damage or resection. The distinction between these two scaffold types enables tailored treatment strategies based on the esophageal defect extent and location [8,9,75,76].

#### 4.1.1. Patch-Type Structure

Patch-type scaffolds are primarily used to repair partial-thickness esophageal defects, restoring specific layers of the esophageal wall, typically the mucosal and muscular layers. These scaffolds are designed for seamless integration with native tissue, promoting cell proliferation, tissue regeneration, and vascularization while maintaining esophageal structural integrity. Owing to their relatively simple geometry, patch-type scaffolds allow for easier implantation and rapid healing, making them ideal for localized esophageal defects.

Hanaro Park et al. developed patch-type scaffolds to reconstruct partial esophageal defects. Two types of patches were fabricated, one using 3D printing technology to fabricate a polycaprolactone (PCL) scaffold in a lattice pattern with dimensions of 1.5 × 2 mm, 150 µm strand diameter, and 500 µm thickness. The second, a polyurethane (PU) scaffold produced using electrospinning by dissolving PU in 20% (w/v) N, N-dimethylformamide and electrospinning it through an 18 G needle at 0.5 mL/h, yielding a PU nanofiber scaffold with a thickness of 150–200 µm. Adipose-derived mesenchymal stem cells (ADSCs) were then seeded onto the scaffolds. To improve cell attachment, the PCL scaffold was coated with Matrigel and seeded with ADSCs at a density of 1 × 10^6^ cells/mL, while the PU nanofiber scaffold was coated with fibronectin (FN) before seeding at the same density. Scaffold efficacy in esophageal reconstruction was assessed using an esophageal defect model in 8-week-old rats. Approximately one-third of the mucosal and muscular layers of the esophagus were excised circumferentially, creating a wedge-shaped defect (1.5 × 2 mm). The defect was then sutured with the prepared scaffolds. The study found that both PCL and PU nanofiber scaffolds promoted muscle regeneration, with the PU nanofiber scaffold additionally enhancing re-epithelialization [77].

Silvia Pisani et al. conducted a study aimed at minimizing structural damage and contamination during long-term culture for esophageal tissue reconstruction while also developing a simple and reproducible protocol for optimal cell seeding on biodegradable patches. They utilized a polylactide-co-polycaprolactone (PLA-PCL) copolymer using molar ratios of 85:15 and 70:30 for polymer patch fabrication. Two techniques—temperature-induced precipitation (TIP) and electrospinning (EL) were employed to produce TIP polymeric films (TIP-Fs) and EL matrices (EL-Ms). The resulting TIP-Fs with a PLA-PCL 85:15 ratio exhibited a smooth and less porous surface, while the PLA-PCL 70:30 ratio displayed large porous structures across the entire patch surface. The fiber diameter of EL-Ms ranged from 700 to 800 nm, demonstrating a random orientation that resulted in an interconnected porous structure owing to fiber entanglement. Subsequently, porcine bone marrow-derived MSCs (p-MSCs) were seeded using two methods—floating and fixation with CellCrown™ (Scaffdex, Temper, Finland). Fixation significantly enhanced cell adhesion and proliferation. Furthermore, TIP-Fs demonstrated a more favorable profile for cell adhesion and proliferation. Although TIP-Fs and EL-Ms exhibited different biodegradation properties, both maintained stability in long-term culture. Optimizing these patches holds potential significance in future tissue regeneration applications (Figure 4A) [76].

Ozkan Cesur et al. developed a bioactive and biodegradable bilayer mesh composed of gelatin and PCL film layers incorporated with fibroblast growth factor (FGF). PCL films were prepared by dissolving PCL in 5% (w/v) dichloromethane and casting the solution in a 10 mm glass Petri dish for solvent evaporation. The resulting films were then aminated by immersion in a 10% hexamethylenediamine-isopropanol solution. Following this, a solution containing 2.5 µg/cm^2^ of FGF was applied to the aminated PCL films and dried to complete the bilayer mesh fabrication. For implantation, a 0.5 × 0.5 cm^2^ semicircular defect was created in the anterior esophageal wall of adult rats and subsequently sutured with the bilayer mesh. The experimental results showed that all rat groups survived the procedure. However, upon sacrificing and examination, localized abscess formation was observed macroscopically in the sham group and the group receiving patches without FGF (FGF (−)) but was absent in the group receiving FGF-loaded patches (FGF (+)). Additionally, the FGF (+) group demonstrated significant epithelial regeneration, a notable increase in collagen content in both the submucosal and muscular layers, and a significant reduction in inflammatory cells by day 28. Burst pressure measurements also demonstrated that the FGF (+) group exhibited significantly higher burst pressure values than that of other groups. This study highlights the importance of FGF in esophageal tissue regeneration (Figure 4B) [78].

Jinfa Qin et al. developed layer-by-layer self-assembly (LBL) structured nanofiber mats as potential esophageal tissue substitutes to reduce postoperative complications. A 12% (w/v, g/mL) N6/SF solution was prepared by dissolving Nylon 6 (N6) and silk fibroin (SF) in a 2:1 weight ratio in hexafluoro isopropanol (HFIP) and electrospun into fibrous mats. Subsequently, the mats were immersed in chitosan (CS) solution, washed, and then immersed in a collagen solution in a repetitive cycle, forming LBL deposition structures up to 15 times. The produced LBL-structured nanofiber mats exhibited antimicrobial properties, achieving over 90% efficacy against Staphylococcus aureus and Escherichia coli. Human esophageal epithelial cells (HEEC) cultured on these structures showed superior cell viability compared to non-LBL structured nanofiber mats, with increased cell survival correlating with a greater number of LBL layers. In animal experiments, wedge-shaped esophageal defects measuring 1.5 × 2 mm were created in rats, and LBL-structured nanofiber mats were sutured into place. The results showed that mats with 15 LBL deposition layers were completely covered by cells and tissues, penetrating deeply between the nanofibers, unlike non-LBL structured mats. Furthermore, immunohistochemical staining for VEGF and CD31 revealed significant vascular regeneration, and increased desmin levels suggested enhanced muscle regeneration. The nanofiber mats developed in this study facilitate esophageal regeneration and offer the potential for reducing postoperative complications [79].

Miji Yeo et al. proposed a method for reconstructing esophageal muscle tissue using advanced cell electrospinning (CE) technology, which encapsulates living cells into fibers for controlled cell alignment. An implantable patch was created by electrospinning PCL into a fibrous mat with suitable mechanical properties, onto which cell-containing bioinks were aligned using CE. The CE scaffolds exhibited greater cellular alignment than non-aligned scaffolds and showed increased expression of ECM components, such as FN, collagen I, and collagen IV. Additionally, the expression of α-SMA, a differentiation marker involved in cellular tension, was significantly enhanced. Subsequently, patches containing mesenchymal stem cell-derived smooth muscle cells (SMCs), referred to as CE-SMC patches, were implanted into a rat model of esophageal defect. A 2 mm biopsy punch was used to excise the mucosal and muscular layers, and a 4 mm diameter circular patch was grafted onto the site. The results showed that control patches without cells exhibited increased immune cell infiltration and tissue thickening, whereas the CE-SMC patches displayed a well-developed muscular layer and granulation tissue. Moreover, the newly formed vasculature was abundant, with significantly elevated SM22α and vimentin levels. These findings indicate that the CE scaffold possesses excellent regenerative potential for esophageal muscle tissue [80].

Yang Luo et al. conducted two studies using aligned nano-ferroferric oxide (Fe_3_O_4_) to induce esophageal muscle cell alignment. In the first study, Fe_3_O_4_ was aligned within a poly (ethylene glycol) (PEG) hydrogel to form a micro-/nano-stripe structure, facilitating cellular orientation and muscle tissue regeneration. In the second study, a gelatin-silk fibroin composite hydrogel containing Fe_3_O_4_ was developed, effectively promoting cell alignment through Fe_3_O_4_ filaments. These studies show the potential of Fe_3_O_4_-based scaffolds in guiding cellular organization and enhancing esophageal muscle tissue repair (Figure 4C) [81,82].

In the first study, Yang Luo et al. developed a stripe-patterned scaffold using poly (ethylene glycol) diacrylate (PEGDA) as the base material fabricated through molding. The scaffold was then coated with an aqueous solution of Fe_3_O_4_ nanoparticles (NPs). A linear magnetic field was applied by positioning magnets at the north (N) and south (S) poles to align the Fe_3_O_4_ NPs. Subsequently, a thin gold layer was deposited onto the aligned Fe_3_O_4_ NPs using ion sputter coating, stabilizing the alignment and creating a modified PEG scaffold with Fe_3_O_4_ micro-/nano-stripes (PEGM). SMCs were cultured on the scaffold to examine their characteristics. The cells exhibited a clear alignment along its structure, as indicated by the organized arrangement of α-SMA and vimentin. The engineered patch was then transplanted into a 4 mm esophageal muscle tissue defect in a rat model. On the first day post-implantation, a significant Ki67 expression was shown, indicating cell proliferation. By day 5, α-SMA expression remained stable, and ACTN2, indicative of smooth muscle contraction, was observed at levels comparable to the surrounding tissue. The implanted patch enhanced cell proliferation and migration, supporting esophageal muscle tissue regeneration. These findings highlight the efficacy of the PEGM scaffold in promoting cellular alignment and tissue repair [81].

In their second study, the authors developed an anisotropic esophageal muscle patch with aligned cells by synthesizing gelatin methacrylate (GelMA) and silk fibroin methacrylate (SFMA) to create a printable hydrogel. Fe_3_O_4_ NPs were incorporated into the hydrogel to induce cell alignment and growth. Bone marrow mesenchymal stem cells (BMSCs) were incorporated into the bioink and printed anisotropically using 3D bioprinting. To fabricate a transplantable scaffold, BMSCs were differentiated into SMCs in vitro using TGF-β1. For implantation, a ~3 × 5 mm defect was created in the rabbit esophagus, and the scaffold was covered and transplanted. By day 9, muscle tissue had fully regenerated, with markers such as α-SMA and connexin 43 expressed at the wound site, indicating functional muscle recovery. Additionally, elevated FN and collagen I expression levels were observed, confirming effective ECM reconstruction in the damaged muscle tissue. These findings suggest that Fe_3_O_4_ NP-incorporated hydrogels represent a promising approach for muscle tissue regeneration therapies [82].

**Figure 4 bioengineering-12-00479-f004:**
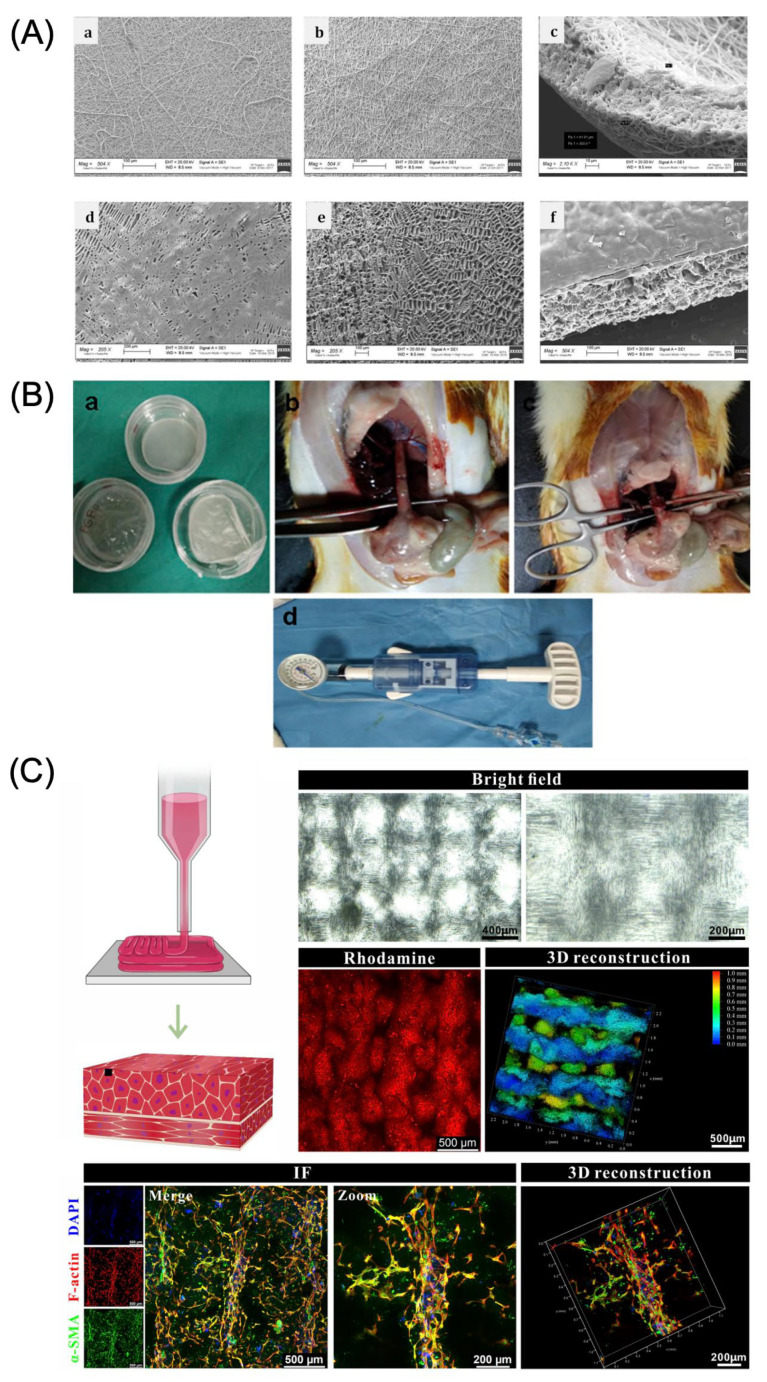
Patch-type scaffolds for esophageal tissue engineering. (**A**) SEM images of electrospun bilayer matrices (EL-Ms) and temperature-induced precipitation films (TIP-Fs). Reproduced under the Creative Commons Attribution 4.0 International license (CC-BY) [76]. (**B**) Design of a study for a bioactive and biodegradable bilayer mesh-based scaffold. Reproduced under the CC-BY 4.0 license [78]. (**C**) Bright-field and fluorescence images of an anisotropic smooth muscle patch fabricated using 3D printing. Reproduced under the CC-BY 4.0 license [82].

#### 4.1.2. Tubular-Type Structure

Tubular-type scaffolds are designed for full-thickness esophageal reconstruction, particularly in cases of circumferential defects or extensive resections. These scaffolds are meticulously engineered to replicate the complex cylindrical architecture of the esophagus, providing the necessary mechanical strength and functional support for effective restoration. To ensure successful integration, tubular scaffolds must withstand peristaltic movements, maintain luminal patency, and promote tissue regeneration while seamlessly integrating with native esophageal tissue.

Gokhan Gundogdu et al. developed a tubular bilayer silk fibroin (BLSF) scaffold as a potential esophageal tissue substitute. The scaffold was fabricated from a silk fibroin solution, extracted and processed from Bombyx mori cocoons, and designed to promote esophageal tissue regeneration. To evaluate its efficacy, the scaffold was implanted into the esophagi of pigs. A 2 cm section of the lower esophagus, 5 cm above the LES, was resected with the vagus nerve preserved, and the BLSF scaffold was implanted in place of the excised tissue. At 5 weeks post-operation, severe stenosis was observed, which was treated successfully with repeated balloon dilations. Histological analysis revealed extensive host tissue regeneration throughout the implanted scaffold. The regenerated mucosa featured a well-vascularized lamina propria lined with stratified squamous epithelium, while the newly formed muscle layer comprised circular and longitudinal layers of both striated skeletal and smooth muscle bundles. This study successfully showed the feasibility of using a cell-free BLSF scaffold for esophageal tissue replacement, highlighting its potential for esophageal tissue engineering applications [83].

Wissam Farhat et al. developed a 3D-bioprinted scaffold for esophageal tissue reconstruction using a polymer blend. The ink, comprising PCL, poly (caprolactone-co-glycolide), PEG, methylcellulose, and glycerol, was processed through a melt–extrusion method to fabricate tubular scaffolds. The printed scaffolds demonstrated mechanical strength similar to that of native esophageal tissue. When sutured to pig esophagi and tested for leaks, the scaffolds withstood pressures of up to 140 mbar, with minimal leakage at the junction between the scaffold and native tissue. Biocompatibility assessments using BALB/3T3 fibroblasts confirmed that sterilized and washed scaffolds were non-toxic. Furthermore, scaffold interactions with TR146 epithelial cells and endothelial progenitor cells (EPCs) were evaluated. TR146 cells exhibited tight adhesion to the scaffold, while EPCs attached and migrated into it, indicating favorable interaction and vascularization potential. Over eight weeks, the scaffolds demonstrated slow biodegradation of about 11–16% in PBS and SGF solutions without swelling. This study highlights the potential of 3D bioprinting to fabricate simple yet high-performance esophageal substitutes for esophageal tissue engineering [84].

Hyoryung Nam et al. employed 3D bioprinting technology to fabricate a porous polymer framework, followed by the production of a 3D artificial esophageal structure using decellularized bioinks derived from porcine mucosal and muscular layers. To facilitate nutrient diffusion and cellular interactions, they employed a dragging technique that rapidly moved the printing nozzle to form pores. This approach produced a flexible, multilayered, free-form tubular construct (MFT) from PCL, replicating the histological composition of the esophagus. Afterward, decellularized bioinks derived from the porcine esophageal mucosa and muscle were printed onto the MFT scaffold. These bioinks retained ECM components while effectively removing cell nuclei and DNA. Cell proliferation analyses revealed significantly higher proliferation in the porous scaffold than in the non-porous scaffold, where dead cells accumulated at the inner–outer layer boundary. Similar results were observed in an in vivo rat intraperitoneal implantation model. Furthermore, gene expression analyses revealed that decellularized bioink derived from porcine esophageal tissue enhanced the functionality of human esophageal epithelial cells (hEECs) and human esophageal smooth muscle cells (hESMCs) within the structure. This study highlights the potential of integrating dECM bioinks with MFT scaffolds for reconstructing circumferential esophageal defects [85].

Tiffany L. Sarrafian et al. investigated the potential of PU scaffolds for esophageal tissue regeneration. The PU scaffolds were fabricated using electrospinning, a technique in which a PU solution was deposited onto a rotating cylindrical collector. In their study, full-thickness esophageal defects measuring 5 or 10 cm were surgically induced in a porcine model, followed by implantation of acellular PU scaffolds. Observation of the recovery process following scaffold implantation shows progressive re-epithelialization, with complete epithelial coverage achieved by 60 days post-operation. However, muscle regeneration was not observed. The authors hypothesize that tissue regeneration might occur through fibrous tissue formation and scaffold contraction. Although the scaffold successfully restores the functionality of food passage, it fails to replicate the native structural architecture of esophageal tissue. Consequently, the authors emphasize the need for further research to address these limitations [86].

Sriya Yeleswarapu et al. utilized 3D printing technology and decellularized hydrogel to fabricate a bioengineered esophageal construct. Using a stereolithography (SLA) system, they first created a tubular mold with 600 µm pores spaced at 400 µm intervals to enhance media and oxygen diffusion during hydrogel culture. Within this mold, they fabricated a dECM hydrogel derived from caprine esophagus muscle, incorporating L929 mouse fibroblasts. After curing the hydrogel, the construct was cultured in media for 7 days, after which the mold was removed, yielding a self-supporting tubular structure. The construct exhibited excellent shape recovery following the application of external force. Cellular functionality was assessed through phalloidin staining, revealing an elongated, spindle-shaped cell morphology. Hypoxia-inducible factor 1-alpha assays show no signs of hypoxic conditions in the inner layers, indicating effective oxygen and nutrient diffusion during cultivation within the SLA-fabricated mold. Additionally, the expression of alpha α-SMA and smooth muscle myosin heavy chain confirms that the dECM hydrogel promotes fibroblast differentiation into myofibroblasts without requiring additional differentiation factors. This study demonstrates the feasibility of fabricating a self-supporting tubular structure from low-mechanical-strength hydrogels using a novel molding technique, presenting promising applications in esophageal tissue engineering [87].

Shijie Qiu et al. developed a composite scaffold by combining decellularized small intestinal submucosa (dSIS) with polylactic-co-glycolic acid (PLGA) to enhance its mechanical properties. The authors stacked four single layers of dSIS and subsequently deposited electrospun PLGA on top. The PLGA nanofibers, with diameters of approximately 600–900 nm, exhibit strong adhesion to the dSIS layers. Primary rat esophageal smooth muscle cells (ESMCs) and esophageal epithelial cells (EECs) were seeded onto the dSIS and PLGA layers, respectively, and cultured for 7 days. After the 7-day incubation period, ESMCs and EECs demonstrated significant proliferation, indicating the non-cytotoxicity of the scaffold. Furthermore, the expression of α-actin and CK-14 was confirmed, indicating the potential of the scaffold to promote smooth muscle formation and epithelial tissue development. This study lays a significant foundation for esophageal mucosal layer reconstruction and the development of a bi-layered tissue-engineered structure [88].

Silvia Pisani et al. developed an implantable esophageal scaffold using electrospinning to fabricate a copolymer blend of poly-L-lactide (PLA) and PCL. Two types of rotating collectors were used to fabricate cylindrical and asymmetrical scaffolds, with the asymmetrical design featuring diameters of 8 and 10 mm optimized for implantation. Permeability testing (180 Da) showed that the asymmetrical scaffold exhibits greater permeability compared to that of the cylindrical scaffold, probably attributed to its thinner structure and larger pore areas at junction sections. Cellularization experiments using p-MSCs showed that the asymmetrical scaffold exhibits better cell viability, attachment, and proliferation at 7 and 14 days compared to that of the cylindrical scaffold, indicating a more favorable environment for cell growth. For implantation, a porcine model was used in which a 2.5 cm circumferential segment of the esophagus was surgically resected, and two suturing techniques—absorbable continuous and detached stitch sutures—were compared. All animals survived for 7 days post-surgery. However, partial detachment and inflammatory responses were observed with the absorbable continuous suture, whereas the detached stitch method showed superior outcomes, with no signs of fistula formation. This study highlights the potential of PLA-PCL electrospun scaffolds as a supportive structure for esophageal repair and provides insights into optimal suturing techniques for implantation based on animal model outcomes [89].

## 5. Animal Models in Esophageal Tissue Engineering Research

### 5.1. Overview of Animal Models for Esophageal Tissue Engineering

Selecting an appropriate animal model for esophageal tissue engineering research is crucial for evaluating the efficacy, safety, and clinical applicability of artificial esophageal grafts. Preclinical studies require animal models that closely replicate the human esophageal environment to accurately evaluate graft integration, functional recovery, and potential postoperative complications. A variety of animal models has been utilized in esophageal regeneration research, each offering distinct advantages and limitations depending on factors such as anatomical size, physiological similarity, immune response, and surgical feasibility. An ideal animal model should closely replicate the anatomical and physiological characteristics of the human esophagus. It should also facilitate the evaluation of immune compatibility, tissue regeneration capacity, and functional recovery following transplantation. Beyond biological relevance, practical considerations—including maintenance costs, ethical concerns, and ease of surgical handling—are critical in selecting the most appropriate model [90,91,92,93,94].

In this section, we explore the applications and characteristics of commonly used animal models in esophageal tissue engineering, focusing on rats, rabbits, and pigs. These models were selected based on specific research objectives and experimental stages, ranging from initial biocompatibility testing of biomaterials to large-scale preclinical studies aimed at clinical translation.

#### 5.1.1. Rat

Rats are among the most widely used small animal models in tissue engineering research owing to their low maintenance costs, ease of handling, and accessibility for genetic manipulation. These characteristics make rats particularly well-suited for early-stage studies, where they are frequently employed to evaluate biomaterial biocompatibility and analyze cellular responses. Conversely, rats serve as an ideal model for the initial performance assessment of novel biomaterials designed for esophageal tissue engineering, including scaffolds and hydrogels [85,95,96,97].

However, rat models have some significant limitations. The primary constraint is their relatively small esophageal size and lumen diameter (approximately 3–5 mm), which creates challenges for surgical manipulation and restricts the application of appropriately sized grafts. Anatomical differences further exacerbate these limitations—notably, the rat esophagus lacks submucosal glands, which are present in human and porcine esophagi and play an essential role in lubrication and mucosal protection. The absence of these key anatomical structures restricts the ability of the rat model to fully replicate the human esophageal environment. Physiological differences also contribute to the limitations of rat models. The peristaltic movements and mucosal characteristics of the rat esophagus differ significantly from those of humans, which can hinder the accurate assessment of functional recovery following graft implantation. From an immunological perspective, rats exhibit different immune responses compared to that in humans, posing challenges when evaluating immune reactivity and graft rejection. In xenotransplantation studies, these differences necessitate careful consideration regarding the use of immunosuppressive agents and the methodologies employed to assess immune responses [92,93,94,98].

#### 5.1.2. Rabbit

Rabbits are widely recognized as suitable small animal models for medium-scale tissue engineering research owing to their larger body size and more developed esophageal structure compared to that of rats. The esophageal dimensions in rabbits are sufficiently large to support partial defect and full-thickness graft studies, making it highly applicable for research involving biomaterials and cell-based therapies. Moreover, rabbits are particularly effective for evaluating immune responses and are frequently used in xenotransplantation and allotransplantation experiments. Another advantage of the rabbit model is its relative ease of surgical handling, which facilitates esophageal transplantation and regeneration studies. Endoscopic evaluations and barium swallow studies can be used to assess the function of the transplanted esophagus, providing valuable quantitative data on post-implantation recovery [92,99,100].

Despite these advantages, rabbit models present some notable limitations. Rabbits exhibit high sensitivity to stress, which can lead to cardiovascular complications or even sudden death owing to environmental changes or external stimuli. To mitigate these risks, careful handling is essential during experiments, and sedatives may be administered when necessary to reduce stress. Anatomically, the rabbit esophagus differs from that of the human esophagus, limiting the direct clinical applicability of certain study findings. For instance, variations in muscular layer thickness and mucosal structure between rabbits and humans require careful consideration when evaluating functional recovery and graft integration. From an immunological standpoint, the rabbit immune system does not fully correspond to that of humans, necessitating careful interpretation of immune response data. This consideration is particularly important in studies assessing immune reactivity and graft rejection. Post-transplant complications are another critical consideration in rabbit models, as rabbits are prone to developing strictures and fistulas at the graft site, which can negatively affect the accuracy of long-term functional assessments. To minimize these risks, precise surgical techniques and comprehensive postoperative care are essential. Implementing effective strategies for preventing or managing such complications is crucial to ensuring reliable and reproducible experimental outcomes [94,100].

#### 5.1.3. Pig

Pigs are widely recognized as the most anatomically and physiologically comparable large-animal model to humans for esophageal tissue engineering research and are extensively utilized in preclinical studies. The porcine esophagus closely resembles the human esophagus in size and thickness, making it highly suitable for full-thickness esophageal reconstruction studies. Additionally, the porcine esophagus features a multilayered squamous epithelium, submucosal glands, and a muscularis propria layer, all of which closely replicate the anatomical structures found in the human esophagus. These similarities make pigs an optimal model for evaluating graft functionality, assessing esophageal regeneration, and measuring functional recovery following transplantation. Physiologically, the porcine esophagus demonstrates peristaltic movements similar to those observed in humans, enabling the application of diagnostic techniques such as endoscopy, barium swallow studies, and esophageal manometry to quantitatively assess post-transplantation esophageal function. These features are critical for accurately evaluating the in vivo performance and functional integration of tissue-engineered grafts. From an immunological perspective, pigs serve as valuable models for xenotransplantation research, as their immune responses closely resemble those of humans. This makes them particularly valuable for assessing immune rejection and inflammatory responses to implanted grafts. Their anatomical size and physiological characteristics make pigs well-suited for testing medical devices and evaluating endoscopic technologies [92,93,94,101,102,103].

Despite these advantages, the use of pig models presents some notable challenges. As large animals, pigs require extensive resources for housing and maintenance, including spacious enclosures, substantial feed, and specialized care, all of which contribute to higher research costs. Surgically, their thick skin and well-developed muscular layers make incisions and suturing more technically demanding compared to that in small animal models. This necessitates advanced surgical expertise and specialized equipment. Furthermore, pigs exhibit a relatively high risk of postoperative complications, including stricture formation and fistula development at the graft site, which can compromise the reliability of experimental outcomes. Minimizing these risks requires precise surgical techniques and careful postoperative management [90,93,101,102,104].

### 5.2. Application of Three-Dimensional Biofabrication in Animal Models for Esophageal Regeneration

Table 3 presents the various scaffold types used for esophageal tissue regeneration and functional restoration across different animal models. Scaffolds designed for esophageal defect reconstruction are primarily categorized into patch and tubular types—each form utilizing different fabrication techniques—such as electrospinning, 3D printing, and decellularization, to replicate the structural and functional properties of native esophageal tissue. These approaches are crucial for replicating the intricate structural and functional characteristics of esophageal tissues.

In animal studies, patch-type scaffolds are predominantly utilized for partial-thickness esophageal defects, where their integration and re-epithelialization are assessed. Studies show that nanofiber-based electrospun patches enhance epithelial coverage, while 3D-printed lattice structures enhance mechanical stability. Conversely, tubular-type scaffolds are primarily used for full-thickness esophageal reconstruction, with assessments focusing on luminal patency, peristaltic function, and long-term tissue integration.

While these approaches demonstrate significant potential, challenges persist in achieving complete functional restoration. Many studies reported high incidences of anastomotic stricture, limited long-term graft viability, and the need for enhanced vascularization. Future advancements should focus on optimizing scaffold composition, enhancing biocompatibility, and incorporating bioactive modifications to enhance seamless tissue regeneration in clinical applications.

## 6. Challenges and Future Perspectives

Esophagectomy remains the gold standard for treating esophageal cancer, offering the most effective method for tumor removal. However, current reconstruction techniques, including gastric pull-up or colonic interposition, pose significant challenges. These approaches often fail to replicate native esophageal peristalsis, leading to swallowing difficulties, gastroesophageal reflux, and impaired digestive function. Postoperative complications such as anastomotic leakage, strictures, and motility disorders significantly affect long-term recovery. While esophagectomy remains essential, the limitations of using other organs for reconstruction highlight the need for alternative approaches that can more effectively restore structural integrity and functional regeneration.

To address these challenges, tissue engineering-based strategies have emerged as a promising alternative. Recent advancements in 3D biofabrication, including bioprinting and electrospinning, have enabled the development of patient-specific esophageal scaffolds designed to restore native tissue properties. These patch- and tubular-type scaffolds have the potential for localized tissue repair and full-thickness esophageal replacement, respectively. However, some challenges persist, particularly in achieving sufficient mechanical strength, facilitating multilayered tissue integration, and ensuring functional adaptation post-implantation. Engineered esophageal constructs must not only support peristaltic motion but also ensure long-term epithelial stability and resist breakdown under physiological stress. To enhance clinical outcomes, advanced scaffold designs must incorporate biomechanical properties that can accommodate dynamic esophageal movements while promoting cell adhesion and tissue remodeling.

To develop bioengineered esophageal constructs capable of restoring structural integrity and functional performance, several key advancements are required. First, optimizing esophageal bioreactors is critical for enhancing scaffold preconditioning before transplantation. These bioreactors create controlled physiological environments, promoting cell viability, differentiation, and mechanical properties, thereby enabling scaffolds to better withstand the dynamic biomechanical forces of the esophagus. Second, ensuring the proper integration of vascular structures is essential for graft survival. Bioengineered esophageal constructs must seamlessly integrate with surrounding tissues to support epithelial regeneration, facilitate smooth muscle function, and minimize immune rejection and fibrosis formation. The integration of vascular structures within an engineered esophageal scaffold is essential for post-implantation adaptation and functional incorporation into host tissue. Additionally, patient-specific modeling through computational simulations can facilitate precision-designed scaffolds tailored to the anatomy and biomechanical properties of an individual. Lastly, establishing standardized preclinical testing protocols is crucial to ensure the safety, functionality, and long-term viability of biofabricated esophageal grafts before clinical application.

Despite notable progress in esophageal tissue engineering, several critical challenges remain before these technologies can be successfully translated into clinical practice. One of the foremost obstacles lies in navigating the complex regulatory landscape [105]. In the United States, tissue-engineered products may be regulated through several distinct pathways established by the Food and Drug Administration (FDA), including Biologics License Applications (BLAs), 510 (k) notifications for lower-risk devices, or Premarket Approval for Class III devices. The specific pathway depends on the product’s classification, and each route involves varying requirements for safety, efficacy, and manufacturing data. The esophagus presents distinct regulatory challenges owing to its complex anatomical structure and dynamic physiological function—particularly its multilayered architecture and coordinated peristaltic motion—which hinder the development of standardized evaluation criteria [9,106]. In current FDA guidance for esophageal prostheses, premarket submissions are required to include detailed data on key performance metrics, such as expansion and compression forces, corrosion resistance, dimensional stability, and biocompatibility. However, these guidelines primarily address conventional devices, and remain insufficient for more advanced tissue-engineered constructs. In particular, there is a lack of clear regulatory benchmarks for evaluating mechanical durability, long-term tissue integration, and functional restoration. To facilitate safe and effective clinical translation, it will be crucial to refine evaluation protocols through interdisciplinary input from both clinical and bioengineering domains.

Beyond regulatory barriers, the limited clinical experience with engineered esophageal constructs remains a significant obstacle to broader adoption [9,107]. Most reported cases have relied on acellular scaffolds or biologically derived patches [107]. For instance, one study demonstrated the repair of large esophageal defects in four patients using an eight-layer, porcine-derived extracellular matrix patch. Within two months, endoscopic examination confirmed complete mucosal coverage, and biopsies revealed the presence of stratified squamous epithelium, indicating the potential for clinical translation [75]. However, despite these encouraging short-term results, the study also underscores the need for long-term outcome data—particularly regarding functional restoration such as peristaltic motion, resistance to reflux, and stable tissue integration. Bridging the translational gap between promising preclinical findings and rigorously controlled human trials remains a critical challenge in the field [9,107].

Equally important is the need to establish clear and comprehensive definitions of success in esophageal regeneration. Current assessments often focus on structural incorporation or epithelialization [9,108]. However, functional restoration—including effective peristalsis, synchronized muscle contractions, mucosal barrier function, and neuromuscular coordination—should be considered essential endpoints in future studies. To meet these challenges, advanced biofabrication approaches are under investigation [68,108,109]. These include 4D printing technologies that allow dynamic shape adaptation, co-culture systems incorporating smooth muscle cells and neurons for enhanced physiological relevance.

While mechanical integration, vascularization, and long-term functionality remain significant challenges, steady progress in scaffold design, bioreactor technology, and bio-fabrication methods continues to improve the feasibility of tissue-engineered esophagi. Moving forward, it will be crucial to standardize evaluation protocols, resolve regulatory uncertainties, and ensure safety and reproducibility in preclinical studies. By addressing these barriers, current and emerging strategies may ultimately pave the way toward effective clinical solutions for esophageal reconstruction.

## Figures and Tables

**Figure 1 bioengineering-12-00479-f001:**
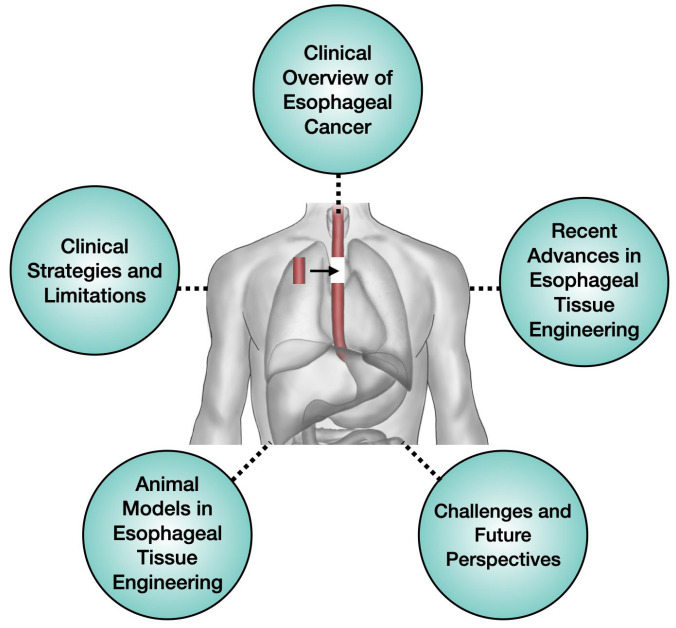
Overview of the main topics covered in this review: clinical overview of esophageal cancer, current clinical strategies and limitations, recent advances in esophageal tissue engineering, animal models for esophageal tissue engineering, and future challenges and perspectives related to esophageal regeneration strategies.

**Figure 2 bioengineering-12-00479-f002:**
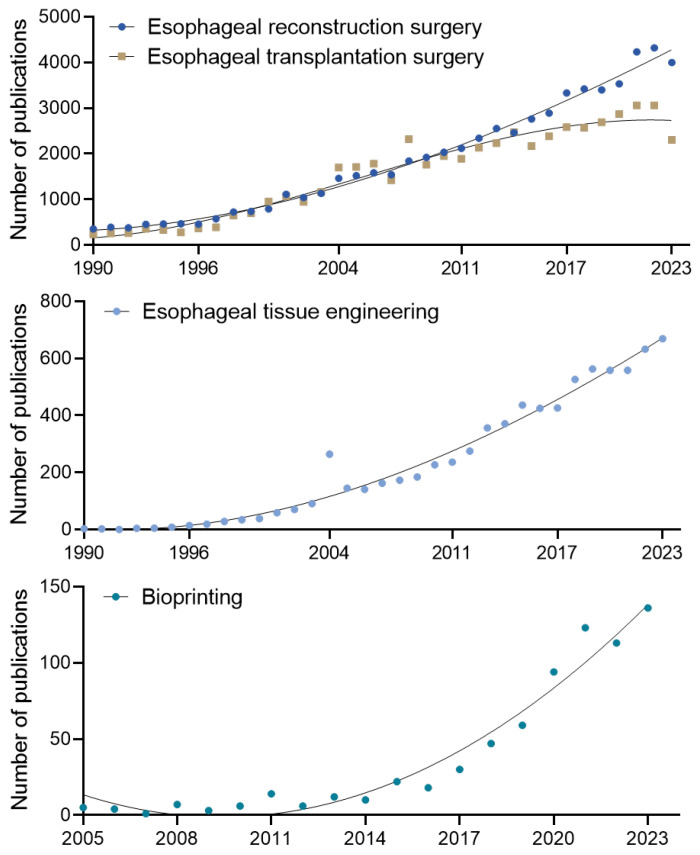
Trends in publication counts from 1990 to 2023 on esophageal regenerative and reconstructive approaches, including esophageal reconstruction surgery, esophageal transplantation surgery, esophageal tissue engineering, and bioprinting. Data were compiled and analyzed by the author using Google Scholar search results.

**Figure 3 bioengineering-12-00479-f003:**
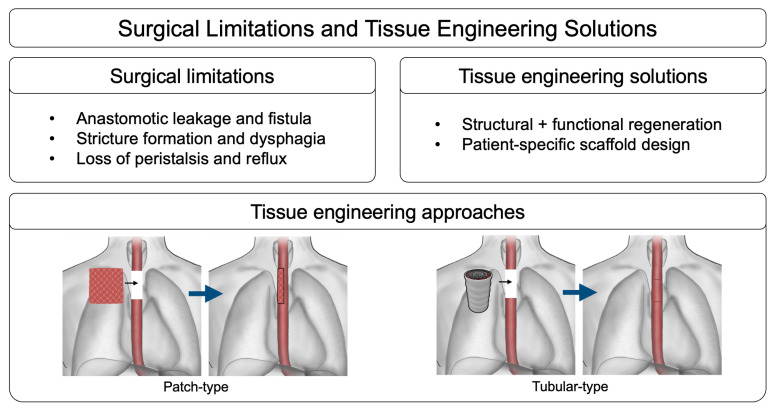
Schematic overview of surgical limitations and tissue engineering solutions for esophageal reconstruction.

**Table 1 bioengineering-12-00479-t001:** American Joint Committee on Cancer (AJCC) Tumor-Node-Metastasis (TNM) Staging System for Esophageal Cancer [43].

Category		Criteria
Primary Tumor (T)	TX	Primary tumors cannot be assessed
	T0	No evidence of a primary tumor
	Tis	High-grade dysplasia (carcinoma in situ)
	T1	Tumor invades the lamina propria, muscularis mucosae, or submucosa
	T1a	Tumor invades the lamina propria or muscularis mucosae
	T1b	Tumor invades the submucosa
	T2	Tumor invades the muscularis propria
	T3	Tumor invades the adventitia
	T4	Tumor invades adjacent structures
	T4a	Resectable tumor invading the pleura, pericardium, or diaphragm
	T4b	Unresectable tumors invading critical adjacent structures, such as the aorta, vertebral body, or trachea
Regional Lymph Nodes (N)	NX	Regional lymph nodes cannot be assessed
	N0	No regional lymph node metastasis
	N1	Metastasis in 1–2 regional lymph nodes
	N2	Metastasis in 3–6 regional lymph nodes
	N3	Metastasis in 7 or more regional lymph nodes
Distant Metastasis (M)	M0	No distant metastasis
	M1	Distant metastasis present

**Table 2 bioengineering-12-00479-t002:** Comparison of the characteristics of major scaffold fabrication techniques and their corresponding strengths and limitations when applied to esophageal tissue engineering.

	Bioprinting	Electrospinning	Mold-Based Fabrication
Description	A layer-by-layer additive manufacturing technique that precisely deposits cell-laden bioinks to construct three-dimensional tissue-like structures.	A fabrication technique that uses a high-voltage electric field to draw polymer solutions into nanofibers, forming scaffolds that mimic the extracellular matrix (ECM).	A traditional technique that involves casting biomaterials or cell-laden hydrogels into pre-designed molds to form tubular or complex tissue structures.
Strengths of 3D Biofabrication in Esophageal Tissue Engineering	- High spatial resolution in the deposition of cells and biomaterials - Capability for fabricating multilayered and anatomically relevant constructs - Integration with patient-specific imaging and digital modeling - Automated and reproducible manufacturing with design flexibility	- Generation of nanofibrous structures mimicking native extracellular matrix - High porosity and surface area facilitating cell attachment and nutrient diffusion - Tunable fiber alignment and diameter for guiding cell orientation - Broad compatibility with natural and synthetic polymers	- Simplicity and scalability of the fabrication method - Efficient generation of cylindrical and luminal geometries - Compatibility with hydrogel systems and hybrid material formulations - High reproducibility and ease of standardization
Limitations of 3D Biofabrication in Esophageal Tissue Engineering	- Insufficient mechanical strength of printed constructs for esophageal function - Reduction in cell viability due to shear stress and prolonged printing duration - Limited availability of bioinks with suitable mechanical and biological properties - High cost and low throughput of the fabrication process	- Inability to construct thick or volumetric tissues with cellular complexity - Difficulty in incorporating live cells during fabrication - Limited mechanical robustness of electrospun scaffolds - Insufficient control over scaffold architecture and layering	- Limited architectural precision and spatial cell distribution control - Inflexibility in accommodating patient-specific anatomical variations - Inability to integrate complex functional tissue interfaces - Risk of uneven material crosslinking or gelation within molds
Advantages for esophageal scaffold	- Accurate replication of esophageal geometry and tissue architecture - Spatial distribution of multiple cell types within defined compartments - Potential for incorporation of vascular and neural components - Customization of scaffold stiffness, porosity, and degradation rate	- Structural similarity to mucosal and basement membrane layers - Enhanced initial cell adhesion and epithelialization potential - Flexibility and conformability to esophageal surfaces - Favorable porosity for oxygen and nutrient exchange	- Structural integrity for maintaining lumen patency - Feasibility for forming thick-walled constructs with simple tubular shapes - Suitability for epithelial and muscle layer seeding - Compatibility with post-fabrication conditioning in bioreactors
Disadvantages for esophageal scaffold	- Inadequate mechanical stability for dynamic peristaltic motion - Risk of construct collapse or deformation after implantation - Heterogeneity in large-volume printing outcomes - Technical complexity in maintaining cell viability and functionality during long fabrication times	- Lack of mechanical strength for supporting esophageal contractility - Unsuitability for smooth muscle regeneration or load-bearing regions - Inability to maintain tubular lumen shape under physiological conditions - Limited applicability for complex multilayered tissue reconstruction	- Lack of resolution for replicating layered histological features - Limited capacity for functional tissue integration (e.g., vascularization) - Inadequate mimicry of dynamic esophageal biomechanics - Constraints in customizing scaffold architecture or incorporating gradients

**Table 3 bioengineering-12-00479-t003:** Animal models in esophageal tissue regeneration study.

Animal Model	Defect Shape	Defect Size	Scaffold Type	Fabrication Techniques (Materials)	Experiment Period	Functional Outcomes	Ref.
Rat	Wedge	1.5 × 2 mm^2^	Patch	Electrospinning (PU) 3D printing (PCL)	4 weeks	PU nanofiber exhibits a tendency to increase re-epithelialization PCL scaffold shows a tendency for more muscle regeneration	[5]
			Patch (sheet)	Electrospinning (nylon6/silk fibroin)	4 weeks	LBL-structured nanofibrous mats display good hydrophilicity, facilitating cell adhesion and proliferation The antibacterial capacity of the mats against *E. coli* and *S. aureus* is >90%	[18]
	Half-circle	0.5 × 0.5 cm^2^	Patch	Solvent casting (PCL) Crosslinking (gelatin, FGF)	4 weeks	A bilayer polymeric mesh containing FGF significantly enhances bioactivity, promoting epithelial regeneration and collagen accumulation	[17]
	Circle	Ø 2 mm	Patch	3D printing (PCL) Cell electrospinning (MSCs, SMCs, alginate, PEO)	2 weeks	CE-SMC patch significantly enhances vascularization, leading to the formation of abundant new blood vessels The CE SMC patch increases the expression of SM22α and vimentin, indicating higher esophageal muscle regeneration	[23]
	Linear	4 mm	Patch	UV molding (PEGDA) Nanoparticle alignment (Fe_3_O_4_) + sputtering (Au)	5 days	Fe_3_O_4_ micro-/nano-stripes, used as alignment inducers within a microchannel-patterned scaffold, promote esophageal muscle tissue regeneration and muscle repair	[24]
Rabbit	Circumferential	1.6 cm	Tubular	Decellularization (pig esophagus)	16 days	A vascularized muscle flap successfully promotes decellularized scaffold anastomoses and neovascularization However, long-term survival remains limited owing to the fragility of the animal model, requiring further testing in larger models	[31]
	Square	~3 × 5 mm^2^	Patch	3D bioprinting (GelMA, SFMA, Fe_3_O_4_, BMSC)	9 days	Hydrogel scaffold supports cell growth and differentiation, aligning BMSCs into SMCs to create a transplantable biomimetic muscle construct It effectively restores smooth muscle structure by enhancing SMC alignment and ECM remodeling	[25]
Pig	Circumferential	4 cm	Cylindrical patch	Decellularization (pig esophagus) QMR	3 months	A QMR-treated scaffold maintains its structural integrity while forming an interconnected network, enhancing cell adhesion and integration BM-MSC-seeded scaffolds enhance esophageal muscle regeneration and reduce inflammation	[32]
		5, 10 cm	Tubular	Electrospinning (PU)	13 months	Although complete esophageal layer formation is not achieved, esophageal healing is observed with stent use A structurally intact tube with patency and no leakage shows significant clinical potential, even if it does not fully replicate the native esophagus	[29]
		2.5 cm	Tubular	Electrospinning (PLA, PCL)	7 days	A PLA-PCL electrospun scaffold provides effective support for promoting esophageal regeneration, as confirmed by preliminary in vitro and in vivo studies	[104]
		2 cm	Tubular	Solvent-casting/salt-leaching (silk fibroin)	3 months	Acellular tubular BLSF implants promote esophageal tissue regeneration, including innervated, vascularized epithelial, and muscular components These grafts show minimal immune reactions and preserve implantation site integrity while, in some cases, enabling oral food consumption	[26] *

* A minipig model was used for this experiment; Abbreviations: PU, polyurethane; PCL, polycaprolactone; FGF, fibroblast growth factor; SMCs, smooth muscle cells; MSCs, mesenchymal stem cells; PEGDA, poly (ethylene glycol) diacrylate; GelMA, gelatin methacrylatep; SFMA, silk fibroin methacrylate; BMSCs, bone marrow mesenchymal stem cells; QMR, quantum molecular resonance; PLA, polylactide; LBL, layer-by-layer; CE, cell electrospinning; ECM, extracellular matrix; BLSF, bilayer silk fibroin.

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
