# Peer review of "Bioengineered Approaches for Esophageal Regeneration: Advancing Esophageal Cancer Therapy"

_bioengineering, 2025, doi:10.3390/bioengineering12050479_

Round 1

Reviewer 1 Report

Comments and Suggestions for Authors

Comments:

  1. It is recommended that the authors provide a schematic drawing to highlight the current limitations in existing surgical approaches and what are some possible solutions to this problem in Section 3.

  1. More information on the different 3D biofabrication techniques should be presented in Section 4.1. For example, what are the different 3D bioprinting techniques in the 2nd paragraph of the Introduction. These techniques include extrusion-based, jetting-based, and vat photopolymerization-based bioprinting. The authors should highlight the advantages and limitations of each bioprinting technique with some relevant references. The authors can refer to some of the papers below:
    1. Extrusion-based
      1. "Extrusion bioprinting: Recent progress, challenges, and future opportunities." Bioprinting21 (2021): e00116.
    2. Jetting-based
      1. "Jetting-based bioprinting: process, dispense physics, and applications." Bio-Design and Manufacturing7, no. 5 (2024): 771-799.
    3. Vat photopolymerization-based
      1. "A review on fabricating tissue scaffolds using vat photopolymerization." Acta biomaterialia74 (2018): 90-111.

  1. The authors should also highlight what are the influence of different bioprinting techniques (extrusion, jetting and vat photopolymerization) on cell viability. What are some of the critical parameters that will affect cell viability? These should be covered in detail within this review paper with relevant references.

  1. Similarly, what are the advantages and limitations of electrospinning and mold-based fabrication?? Are these processes compatible with living cells or cell-seeding process is required after fabrication?

  1. The authors should provide a concise table summarizing the studies in Section 4 (bioprinting, electrospinning and mold-based fabrication).

  1. What about the regulatory challenges?? These should be discussed in Section 6.

Reviewer 2 Report

Comments and Suggestions for Authors

The article is a review article – devoted to advances in esophageal tissue engineering with an emphasis on clinical perspectives, technological innovations and emerging issues.
The review article is well written and allows the reader to understand the extensive experimental results. I would like the authors to make the figures larger, as it is very difficult to read small inscriptions. I think the article can be recommended for publication after making minor edits to the text of the article.

The following comments are made to the text of the article:

1. Line 84: the following sentence should be rewritten: “The findings could highlight the latest promising research findings necessary to address key barriers and facilitate future clinical adoption”
2. Figure 3 should be rewritten – all inscriptions should be enlarged.
3. The conclusion lacks review data on possible clinical studies of the developed 3D models. Are there any such publications?

Round 2

Reviewer 1 Report

Comments and Suggestions for Authors

The authors have addressed all the comments, the revised manuscript can be accepted.